# MicroRNA Expression in Extracellular Vesicles from Nasal Lavage Fluid in Chronic Rhinosinusitis

**DOI:** 10.3390/biomedicines9050471

**Published:** 2021-04-26

**Authors:** Seungbin Cha, Eun-Hye Seo, Seung Hyun Lee, Kyung Soo Kim, Chung-Sik Oh, Jong-Seok Moon, Jin Kook Kim

**Affiliations:** 1Department of Infection and Immunology, Konkuk University School of Medicine, Seoul 05030, Korea; sbanny91@gmail.com (S.C.); shlee@kku.ac.kr (S.H.L.); 2Research Institute of Medical Science, Konkuk University School of Medicine, Seoul 05030, Korea; gmreo@naver.com (E.-H.S.); ohcsik@naver.com (C.-S.O.); 3Department of Otorhinolaryngology-Head and Neck Surgery, Chung-Ang University College of Medicine, Seoul 06973, Korea; entkks@cau.ac.kr; 4Department of Anesthesiology and Pain Medicine, Konkuk University School of Medicine, Konkuk University Medical Center, Seoul 05030, Korea; 5Department of Integrated Biomedical Science, Soonchunhyang Institute of Medi-bio Science (SIMS), Soonchunhyang University, Cheonan-si 31151, Korea; jongseok81@sch.ac.kr; 6Departments of Otorhinolaryngology-Head & Neck Surgery, Konkuk University School of Medicine, Konkuk University Medical Center, Seoul 05030, Korea

**Keywords:** sinusitis, nasal lavage fluid, extracellular vesicles, microarray analysis, microRNAs

## Abstract

Extracellular vesicles (EVs) are nanovesicles of endocytic origin released by cells and found in human bodily fluids. EVs contain both mRNA and microRNA (miRNA), which can be shuttled between cells, indicating their role in cell communication. This study investigated whether nasal secretions contain EVs and whether these EVs contain RNA. EVs were isolated from nasal lavage fluid (NLF) using sequential centrifugation. EVs were characterized and EV sizes were identified by transmission electron microscopy (TEM). In addition, EV miRNA expression was different in the chronic rhinosinusitis without nasal polyp (CRSsNP) and chronic rhinosinusitis with nasal polyp (CRSwNP) groups. The Kyoto encyclopedia gene and genome database (KEGG) database was used to identify pathways associated with changed miRNAs in each analysis group. Twelve miRNAs were differentially expressed in NLF-EVs of CRS patients versus HCs. In addition, eight miRNAs were differentially expressed in NLF-EVs of CRSwNP versus CRSsNP patients. The mucin-type O-glycan biosynthesis was a high-ranked predicted pathway in CRS patients versus healthy controls (HCs), and the Transforming growth factor beta (TGF-β) signaling pathway was a high-ranked predicted pathway in CRSwNP versus CRSsNP patients. We demonstrated the presence of and differences in NLF-EV miRNAs between CRS patients and HCs. These findings open up a broad and novel area of research on CRS pathophysiology as driven by miRNA cell communication.

## 1. Introduction

Chronic rhinosinusitis (CRS) is characterized by chronic inflammation of the sinonasal mucosa. Chronic rhinosinusitis without nasal polyps (CRSsNP) and chronic rhinosinusitis with nasal polyps (CRSwNP) are two different CRS phenotypes. Advances in our understanding of CRS pathophysiology have led to adoption of the endotype paradigm of disease characterization. The main goal of CRS research is to understand its etiopathology and significant prognostic information for personalized CRS treatment. Identifying pathways that allow CRS to be expressed is fundamental to improving preventative strategies, developing diagnostic tools, and designing therapies.

Extracellular vesicles (EVs), or exosomes, are present in most bodily fluids, including the nasal mucus [1], and their composition differs by cellular origin. Vesicles in the extracellular space are derived from EVs and from ectosomes (shedding vesicles). EVs contain microRNA (miRNA) and messenger RNA (mRNA) (exosomal shuttle RNA), which are transferable between cells, illustrating their role in cell commutation. The presence of RNA has been confirmed in EVs from the saliva, plasma, breast milk, and nasal lavage fluid (NLF) [2].

The mRNA, microRNA, and proteins of EVs in various bodily fluids are being investigated for their potential use as biomarkers of disease. EVs in plasma and urine differ in their protein and RNA content in cancer patients relative to healthy controls (HCs) [3]. Nocera et al. reported that exosomal P-glycoprotein (P-gp), an efflux pump that drives type-2 helper T-cell inflammation in CRSwNP, is present and significantly enriched in CRSwNP EVs relative to HCs [4]. Wu et al. reported the miRNA content of EVs in allergic rhinitis (AR) and found that EV miRNA expression is altered in AR and that differentially expressed miRNAs appear to be involved in AR development [1]. Xuan et al. reported that miRNA profiles from the nasal mucosa EVs of CRSwNP patients are completely different from those of HCs [5]. However, there are no studies on the comparison of NFL EVs and miRNA profiles in CRS patients and HCs.

This study investigated the presence of EVs in the NLF of CRS patients and compared it to HCs. In addition, miRNA profiles from EVs and relevant biological pathways were evaluated in CRS patients and HCs.

## 2. Materials and Methods

### 2.1. Human Subjects

We conducted a study on CRS patients who visited Konkuk University Hospital, Korea, from 2019 to 2020. All patients provided prior written informed consent, and the study was approved by the institutional review board of Konkuk University Medical Center (KUMC 2019-12-012-008 approved on 18 December 2019).

NLF was collected from the HCs during septoplasty and the CRS patients during endoscopic sinus surgery under general anesthesia. CRS patients were classified according to the presence or absence of nasal polyps into CRSsNP or CRSwNP patients, respectively. AR was confirmed by the skin-prick multiple-allergen simultaneous test (MAST), and negative results were confirmed by serum allergen-specific immunoglobulin E (IgE). The JESREC score criterion was used for patient classification because it is suitable for the clinical classification method according to the degree of peripheral blood eosinophilia in East Asian patients. The JESREC scoring method has been previously described [6]. Briefly, we scored 3 points for bilateral disease sites, 2 points for polyps, and 2 points for the ethmoid ≥ maxillary in the CT shadow. Additionally, for the peripheral blood eosinophilia, 4 points were scored for 2 <≤ 5% eosinophils of peripheral blood, 8 points for 5 <≤ 10%, and 10 points for >10%. Finally, a total of 11 points were defined as the cutoff value of the JESREC score. If it was 11 points or higher, it was diagnosed as ECRS, and scores below 10 were diagnosed as non-ECRS. The study cohort comprised total 312 participants (Appendix A). The cohort was further divided into cohort 1 and cohort 2 according to the number of recruitments by period. Cohort 1 comprised 44 CRSwNP patients, 108 CRSsNP patients, and 77 HCs and was used to characterize EVs in the NLF (Appendix A). Cohort 2 comprised 18 CRSwNP patients, 28 CRSsNP patients, and 37 HCs and was used to analyze the NLF. From all of these, we selected 7 CRSwNP patients, 8 CRSsNP patients, and 7 HCs as representative groups and analyzed them for miRNAs (Table 1).

### 2.2. NLF Collection

The NLF was collected from the participants. Briefly, a 3-way catheter was inserted into the nasal cavity to the end for nasal irrigation, ballooned in the posterior choana area, pulled forward so that the irrigation saline did not fall down the neck, and then ballooned in the nostril area. Using a new syringe, normal saline was injected into the nasal cavity through the foley and regurgitated to collect the irrigation sample in a screw bottle. Irrigation was repeated until the total irrigation sample was at least 10 mL.

### 2.3. Isolation of EVs from the NLF

Recovery and dilution of the NLF were performed using RNAse-free ice-cold phosphate-buffered saline (PBS). Immediately after sampling 10 mL of the NLF, it was centrifuged at 300× *g* for 10 min at 4 °C, at 2000× *g* for 20 min, and at 16,500× *g* for 50 min. The pellet containing the cell debris was discarded, and the supernatant was removed. Next, the supernatant was centrifuged again at 120,000× *g* for 60 min, and the pellet was vortexed with 1 mL of ice-cold PBS to collect the sample.

### 2.4. EV Characterization

EV samples were stained with fluorescein isothiocyanate (FITC) mouse antihuman CD9 (555371) and PerCP-Cy™5.5 mouse antihuman CD63 antibodies (565426) from BD Biosciences (San Jose, CA, USA) for 30 min at room temperature under dark lighting. EVs were acquired using an Accuri C6 flow cytometer (BD Biosciences), and analysis was performed using FlowJo software (Treestar, Inc., San Carlos, CA, USA). EVs stained in the same way were also analyzed using confocal microscopy with a Carl Zeiss microscope (LSM900, Zeiss, Oberkochen, Germany). Image analysis was performed using ZEN3.0 (blue edition) software.

Transmission electron microscopy (TEM) analysis was performed by fixing EV pellets with 4% paraformaldehyde (PFA) and then by negative-staining them with 1% uranyl acetate. The equipment used for analyzing EVs mounted on the grid was Tecnai™ G2-F20 (FEI, OR, USA).

### 2.5. MicroRNA Isolation from EVs

Total RNA was extracted from 400 µL of EV using the Promega Maxwell^®^ RSC miRNA Tissue Kit (AS1460; Promega Corporation, Fitchburg, WI, USA) and Maxwell^®^ RSC instrument (Promega Corporation, Madison, WI, USA) according to the manufacturer’s instructions. The concentration and purity of the extracted total RNA were determined using Nanodrop™ ND2000 (Thermo Fisher Scientific, Inc., MA, USA) and the AATI Fragment Analyzer (Advanced Analytical Technologies, CA, USA) to determine the miRNA content in the sample. 

### 2.6. Nanostring nCounter System MiRNA Assay

The expression profile of 798 human miRNAs was evaluated using Human v3 miRNA Assay Kit (NanoString Technologies, Seattle, WA, USA) according to the manufacturer’s instructions. The high sensitivity and specificity of this method are well established. The expression profile was evaluated on EV miRNA from NLF, starting with 100 ng of total RNA. To investigate the relative abundance of the detected EV miRNA, after normalizing the sample, we tried to further analyze the differentially expressed miRNA using the fold-change (FC) value of each group. The data were statistically analyzed using nSolver 4.0 software (NanoString Technologies) (Appendix A) and subjected to volcano plots, agglomerative clusters, and heat maps using the Bioconductor R package. 

### 2.7. Prediction of Target Genes and Associated Pathways

To predict target genes for the miRNA and related biological molecular pathways through microT-CDS (v5.0) and the Kyoto Encyclopedia Gene and Genome database (KEGG) pathways, we used DIANA-mirPath software v.3.0, a web-based application (http://diana.imis.athena-innovation.gr/DianaTools/index.php. Accessed on 5 March 2021) [7].

For the analysis, we used miRNAs from the comparison group with FC ≥ 2 and *p* < 0.05. The results include a list of predictable pathways corresponding to the miRNA list and a list of genes involved in the miRNA.

### 2.8. Statistical Analysis

The data were analyzed using GraphPad Prism version 4.0 (GraphPad Software, Inc., San Diego, CA, USA) and provided as the mean ± standard deviation (SD) or median ± interquartile range (IQR). Significant differences between groups were indicated at *p* < 0.05 after performing Student’s *t*-test or analysis of variance (ANOVA) with Tukey’s multiple comparison test.

## 3. Results

### 3.1. Clinical Characteristics of the Study Participants 

Our overall experimental process is shown in Appendix A. Appendix A shows the demographics of cohort 1, and Table 1 shows the demographics of cohort 2. There were no significant differences between CRS patients and HCs with respect to the mean age, sex, prevalence of asthma, environmental allergies, and JESREC score [6].

### 3.2. Phenotypic Characterization of NLF-EVs

The presence of EVs in NLF samples was investigated according to previous studies and the guidelines of the International Society of Extracellular Vesicles [8]. First, isolated EVs were analyzed by flow cytometry (FCM) with antibodies against the documented EV markers CD9 and CD63 (Appendix A). FCM showed a significant increase in CD63+ or both CD9+ and CD63+ NLF-EVs of CRS patients compared to HCs (*p* = 0.1026 for CD9+ EVs; *** *p* < 0.0002 for CD63+ EVs; *** *p* = 0.0007 for CD9+/CD63+ EVs) but no significant difference between CRSwNP and CRSsNP patients. Little is known about EV concentration in the NLF, but the EV concentration in bronchoalveolar lavage fluid (BALF) can change in lung cancer patients [9]. There were no significant changes in the EV concentration according to sex or age.

Confocal and electron microscopy (EM) of the isolated EVs showed, respectively, CD9+/CD63+ EVs (Appendix A) and a clear, spheroid morphology in a size range of 70–250 nm; the median size was 124.57 nm (IQR = 93.78–157.14) (Figure 1 and Table 2).

Next, we quantitated the total RNA amount harvested from EVs using the AATI Fragment Analyzer. The average amounts of total RNA were 12.75 ng/µL in HCs, 51.52 ng/µL in CRSwNP patients, and 29.39 ng/µL in CRSsNP patients, indicating that CRS induces an increase in the total EV RNA (Appendix A). The RNA species were 50–100 base pairs in all groups, which is consistent with the miRNA size (data not shown), indicating that the EV morphology did not change but that the amount of EVs and EV RNA, especially miRNA, significantly increased in CRS patients’ NLF.

### 3.3. MicroRNAs Were Differentially Expressed in NLF-EVs of CRS Patients

We profiled the expression of 798 miRNAs in NLF-EVs of the CRSwNP, CRSsNP, and HC groups (Table 1 and Appendix A). Volcano plots were applied between the groups to identify statistically significant differentially expressed miRNAs (Figure 2A). Five upregulated and seven downregulated miRNAs (|FC| > 2 and cutoff *p* ≤ 0.05) were differentially expressed in the NLF-EVs of CRS patients compared to HCs, while eight upregulated miRNAs (|FC| > 2 and cutoff *p* ≤ 0.05) were differentially expressed in the NLF-EVs of CRSwNP versus CRSsNP patients. Using heat map analysis, it was possible to observe global differences in the gene expression profile (green (down) and red (up)) between the CRS patients and HCs, as well as the most representative clustering groups of samples and genes for each patient (Figure 2B). Interestingly, in the CRSwNP versus CRSsNP comparison, some patients were grouped in distinctive clusters and demonstrated a similar expression pattern of miRNAs. In contrast, in the CRS versus HC comparison, some patients were not observed in distinctive clusters but still demonstrated a similar expression pattern of miRNAs. Gender, age, and smoking status showed no obvious influence on the miRNA expression level.

### 3.4. Gene Ontology and Pathway Classification of In Silico Analysis of MiRNA Target Genes in CRS vs. Healthy NLF-EVs

In individual NLF-EVs of CRS patients, miR-1285-3p (FC_miR-1285-3p_ = 2.85; *p* = 0.01) significantly increased while miR-671-3p and miR-15a-5p (FC_miR671-3p_ = −2.19; *p* = 0.01 and FC^miR-15a-5p^ = −2.17; *p* = 0.001) significantly decreased compared to HCs (Figure 3A).

The results of the DIANA-mirPath tool using the miRNA signature showed that 46 KEGG biological processes analyzed in microT-CDS were significantly enriched between miRNAs differentially expressed in NLF-EVs of CRS patients compared to HCs (*p* < 0.05, false discovery rate (FDR) correction). We excluded the list of pathways related to cancer immunity and plotted the main pathways as fold-enrichment/*p*-values and gene count/*p*-values. Of these, mucin-type O-glycan biosynthesis (*p* = 7.31 × 10^−5^), Hippo signaling pathway (*p* = 1.09 × 10^−4^), FoxO signaling pathway ( *p* = 3.62 × 10^−4^), PI3K-Akt signaling pathway (*p* = 3.75 × 10^−3^), focal adhesion (*p* = 6.13 × 10^−3^), adherens junction (*p* = 1.49 × 10^−2^), Rap1 signaling pathway (*p* = 1.53 × 10^−2^), and transforming growth factor-beta (TGF-β) signaling pathway (*p* = 2.21 × 10^−2^) were the most prominent pathways rich in quantile with differential EV miRNA patterns, suggesting that these biological pathways are involved in CRS development (Figure 3B,C). The target location of miRNA corresponding to mucin-type O-glycan biosynthesis is summarized in Figure 3D [10].

### 3.5. Gene Ontology and Pathway Classification of in Silico Analysis of MiRNA Target Genes in CRSsNP vs. CRSwNP NLF-EVs

The results showed that miR-1290, miR-548q, miR-548l, miR-1254, miR-890, and miR-548aa/548t (FC_miR-1290_ = 2.24, *p* = 0.03; FC_miR-548q_ = 2.03, *p* = 0.01; FC_miR-548l_ = 5.53, *p* = 0.02; FC_miR-1254_ = 2.00, *p* = 0.002; FC_miR-890_ = 2.12, *p* = 0.03; and FC_miR-548aa/548t_ = 2.19, *p* = 0.01) were significantly increased in NLF-EVs of CRSwNP compared to CRSsNP patients (Figure 4A).

In addition, 43 KEGG biological processes were significantly enriched between miRNAs differentially expressed in the NLF-EVs of CRSsNP compared to CRSwNP patients (*p* < 0.05, FDR correction). Excluding the list of pathways related to cancer immunity, the main pathways in Figure 4 were plotted as fold-enrichment/*p*-values and gene count/*p*-values in CRSsNP compared to CRSwNP patients, respectively. Of these, the TGF-β signaling pathway (*p* = 8.97 × 10^−7^), Hippo signaling pathway (*p* = 8.97 × 10^−7^), FoxO signaling pathway (*p* = 1.66 × 10^−4^), mucin-type O-glycan biosynthesis (*p* = 2.37 × 10^−4^), adherens junction (*p* = 1.29 × 10^−3^), ErbB signaling pathway (*p* = 1.30× 10^−3^), Rap1 signaling pathway (*p* = 1.30 × 10^−3^), gap junction (*p* = 5.95 × 10^−3^), tight junction (*p* = 1.64 × 10^−2^), and estrogen signaling pathway (*p* = 1.97 × 10^−2^) were the most prominent pathways rich in quantile with differential EV miRNA patterns, suggesting that these biological pathways are involved in the presence or absence of polyps in CRS (Figure 4B,C). The target location of miRNA corresponding to the TGF-β signaling pathway is summarized in Figure 4D [11]. 

### 3.6. Functional Validation of Representative MiRNA Analysis Results

Table 3 summarizes the representative miRNA, target gene, and estimated related disease results that showed differences in expression identified through in silico analysis. Interestingly, most genes were found to be associated with inflammation, disease development, and fibrosis. Assessment of the genes targeted by miR-23a-3p, miR-15a-5p, miR-223-3p, and miR-1285-3p involved in the mucin-type O-glycan biosynthetic signaling pathway predicted representatively the MET proto-oncogene (MET), epidermal growth factor receptor (EGFR), beta-actin (ACTB), transforming growth factor beta receptor (TGF-βR), and poly(ADP-ribose) polymerase 1 (PARP1). The previous studies shown in Table 3 showed each miRNA’s role in disease progression, fibrosis, and inflammation, which was consistent with our results with regard to inflammatory diseases, but opposite results were also confirmed in some references denoted with asterisks [12,13,14,15,16,17,18,19,20,21,22,23].

Assessment of genes targeted by miR-548q, miR-1290, and miR-548l involved in the TGF-β signaling pathway predicted representatively small mothers against decapentaplegic (SMAD), TGF-β, and mitogen-activated protein kinase (MAPK). The previous studies in Table 3 showed each miRNA’s role in disease progression, inflammation, and acute disease [24,25,26,27,28,29,30,31], which was consistent with our results with regard to inflammatory diseases, but opposite results were also confirmed in some references denoted with asterisks [22,23,26].

## 4. Discussion

CRS is a heterogenous group of diseases characterized by chronic inflammation of the nasal and paranasal mucosa [32,33]. However, the mechanism underlying CRS development is unclear. The recent focus on CRS is an important step toward understanding the various inflammatory profiles of CRS patients and may ultimately enable individualized treatment protocols. This study reported that a phenotype group of EV miRNA expression is altered in CRS and that differentially expressed miRNAs appear to be involved in the CRS inflammatory profile. Our findings may lead to a better understanding of the role of EV miRNAs identified in CRS pathogenesis.

EVs are released from cells or directly from the plasma membrane into the peripheral circulatory system in the form of multivesicular bodies with a plasma membrane [3]. Knowing the origin of EVs would be helpful in assessing the importance and function of modified miRNAs in CRS. Numerous researchers have actively investigated how EVs merge with and release their contents into cells that are distant from their cells of origin and how they may influence processes in the recipient cells. By transferring an miRNA from one cell to another, EVs can play a functional role in adaptive immune mediation in some cells of the immune system, such as dendritic cells and B cells [34,35].

Increasing evidence supports the importance of EV miRNA regulation in CRS development, but the underlying mechanism is poorly defined. The prediction and identification of miRNA target genes provides an experimental basis for further study of miRNA regulatory mechanisms. We used bioinformatics methods based on sequence similarity between target genes and miRNA to predict potential target genes. We revealed alterations in the EV miRNA profile of CRS patients, which suggested intrinsic dysregulation of EV content in CRS. The characteristics of EVs in the NLF depend on patients’ demographics classified by the presence of absence of nasal polyps of CRS (Table 1 and Appendix A). EV separation methods include ultracentrifugation (UCF), size-exclusion chromatography, and separation using polymers. We used UCF described by Théry et al., which was compared to the commercially available precipitation method (ExoQuick; System Biosciences, Palo Alto, CA). The isolation of EVs from the NLF provides higher exosome yields and higher protein purity by UCF [4,36].

The identification of size and surface molecular markers is typically used for EV characterization. A combination of several methods, such as dynamic light scattering, nanoparticle tracking analysis (NTA), EM, and FCM, is typically used to determine the EV size. A recent study used FCM to identify and characterize membrane surface markers of EVs, such as CD9 and CD63 [1,37]. Görgens et al. reported that Accuri C6 FCM (BD Biosciences) provides the sizes of cells and particles using the difference between the collection angles of forward scatter (FSC) versus side scatter (SSC) of EVs for which the size is less than 200 nm [37] and is stable for analysis of the EV size. We used FCM and double-checked results with fluorescence immunohistochemistry.

The commonly known EV markers are CD9, CD63, and CD81. Lässer was the first to confirm that the NLF contains EVs and to identify exosome surface markers CD9, CD63, and CD81 using FCM [2]. In addition, since the endoplasmic reticulum (ER) and the plasma membrane, as multiple cell compartments including multivesicular bodies (MVBs), can release vesicles, to confirm the intracellular origin and characteristics of EVs, ER marker Tsg101 and MVB marker calnexin were used in combination.

Nocera compared the amounts of CD63+ EVs in the NLF between the CRSwNP and control groups using an enzyme-linked immunosorbent assay, but there was no significant difference between total CD63+ and mean CD63+ concentrations between the CRSwNP and control groups [4]. In contrast, Qazi confirmed an increase in EV production in sarcoidosis patients by analyzing the correlation between the concentration of cells and EV proteins in EVs in the BALF. They reported that increased infiltration in lung cells; cell activation status; or primary defects in patients, such as tuberculosis, may contribute to the EV profile of sarcoidosis and the inflammatory response of sarcoidosis due to their ability to stimulate cytokine responses [38].

We used EM to visualize the size and shape of EVs. Kesimer analyzed EVs from airway epithelial cells. EVs have a dark-gray color in the center of their structure as a general characteristic on TEM [39]. Lässer and Nocera used TEM to discover structures corresponding to the size and spheroid morphology in the EV fraction of the NLF from 30 to 150 nm [2,4]. Measurement of the EV size varies depending on the isolation and measurement method, but EVs in human fluids (plasma, urine) are generally 20–300 nm [40,41,42].

The EV concentration reflects the balance between their secretion and internalization. EVs in the NLF may interact with nasal epithelial cells (NECs) and circulating cells. In our study, the EV miRNA amount in the NLF of CRS patients differed from that of HCs. Reports on the correlation between EV miRNA levels in body fluids and disease continue to emerge. However, in recent reports, such as “One Airway, One Disease,” only the sites of occurrence are different, and Huse et al. regarded them as the same disease with a correlation between nasal diseases and respiratory diseases [43]. The investigator noted that patients with more severe rhinitis were much more likely to have “moderate to severe asthma” (60.2% vs. 51.2%, respectively). However, spirometry values and the incidence of nasal polyps were not mentioned. Although controversy remains regarding this, Sabine used NLF for research on asthma and demonstrated that the symptoms of asthma are transmitted to the EV of NLF through EVs secreted from human lung epithelial cells. They showed that the information contained in NLF and some miRNAs in the airway fluid (e.g., BALF) are significantly correlated with airway disease diagnosis factors, such as lung function (FEV1) [44]. Therefore, we investigated the internalization of EV miRNA levels by the human NLF in serial results.

MiRNAs are important posttranscriptional regulators of gene expression that control CRS development. Dysregulation of miRNA-mediated mechanisms is an important pathological factor in chronic inflammatory-related diseases. We identified twelve differentially expressed miRNAs with more than twofold changes, including seven upregulated miRNAs (miR-15a-5p, miR-671-3p, miR-142-3p, miR-25-3p, miR-223-3p, miR-23a-3p, and miR-941) and five downregulated miRNAs(miR-1285-3p, miR-1469, miR-450a-1-3p, miR-650, and miR-664b-5p) in the NLF-EVs of CRS patients versus HCs. In addition, eight upregulated miRNAs (miR-890, miR-519a-5p, miR-1254, miR-548t-3p, miR-1290, miR-548l, miR-376c-5p, and miR-548q) were differentially expressed in the NLF-EVs of CRSwNP versus CRSsNP patients (Figure 2).

Xuan et al. used microarray analysis to compare the miRNA expression profiles in the nasal mucosa of CRS patients and HCs [5]. They identified 24 differentially expressed miRNAs with more than twofold changes, including 5 upregulated miRNAs (miR-210-5p, miR-3178, miR-585-3p, miR-3146, and miR-320e) and 19 downregulated miRNAs (miR-32-3p, miR-1299, miR-3196, miR-3924, miR-548e-3p, miR-3184-5p, miR-375, miR-23a-5p, miR-377-5p, miR-574-5p, miR-3149, miR-500a-5p, miR-125b-2-3p, miR-1914-5p, miR-532-3p, miR-612, miR-1298-5p, miR-1226-3p, and miR-688-3p). Xia measured the expression levels of seven miRNAs differentially expressed in CRS and considered them to be associated with inflammatory or immune responses using quantitative reverse transcription polymerase chain reaction [45]. CRSwNP patients had increased miR-125b, miR-155, and miR-146a but decreased miR-26b, miR-92a, and miR-181b in the nasal mucosa compared to CRSsNP patients. We performed a pilot study of microarray analysis using tissue samples from 8 patients, from whom the NLF-EV samples were obtained (data not shown). Among the miRNAs from nasal tissue, three miRNAs (miR-223-3p, miR-25-3p, miR-671-3p, and miR-664b-5p) identified the same results as that in the miRNAs of NLF-EV. In the same comparison groups as in Xuan’s and Xia’s studies, we confirmed the results in 4 miRNAs (miR-155, miR-375, miR-671-3p, and miR-92a-3p). It is confusing that, between Xuan’s and Xia’s studies, we can find more discrepancies than consistencies in CRS miRNA profiles. Although studying the miRNA expression profiles in tissues can explain gene regulation in the diseased mucosa, there can be a significant degree in cellular heterogeneity across individual biopsies. The underlying reason is not clear, but it may be because of organizational heterogeneity. 

Nevertheless, the type of miRNA derived in this study was different from those in other studies due to the limitation of not being able to completely exclude individual biopsies and cellular heterogeneity and to identify environmental factors. Regardless, the obtained results were included in the final mechanism analysis. The results of the KEGG biological processes using DIANA-mirPath in Xuan’s study showed that upregulated miRNAs were involved in thyroid hormone synthesis, mucin-type O-glycan biosynthesis, and the estrogen signaling pathway and that downregulated miRNAs were involved in the TGF-β signaling and gap junction pathways [5].

In Zhang’s study, CRSsNP and eosinophilic CRSwNP patients showed distinct miRNA expression profiles. To identify putative target genes, they performed bioinformatics analysis using miRanda, TargetScan, and PicTar algorithms. They selected miR-125b genes that were predicted by all three algorithms. Among the target genes, *4E-BP1* is an important downstream element of the phosphatidylinositol-3 kinases/serine/threonine kinase Akt/mammalian target of the rapamycin pathway, which critically affects broad aspects of cellular functions [46].

In this study, we selected mucin-type O-glycan biosynthesis as the representative mechanism for CRS development. Three predicted upregulated miRNAs (miR-223a-3p, miR-23a-3p, and miR-15a-5p) and two downregulated miRNAs (miR-1285-3p and miR-450a-1-3p) were involved in the regulation of mucus production in CRS NLF-EVs by targeting GALNTL2 and B4GALT, which may influence the transference of sialic acids to O-glycan.

Reports on mucin-type O-glycan biosynthesis in CRS were summarized in Xuan’s study by analyzing the miRNA factor that induces CRSwNP in nasal polyp tissue. In a healthy person, mucus assembles into a protective gel-like barrier to lubricate and hydrate the epithelial layer as well as to clear out dust and exogenous luminal insults. However, the authors found the accumulation of sticky and thick mucus that may create a favorable growth environment for pathogens in CRS patients. In this biological pathway, the main gel-forming components of mucus covering the surface of the nasal mucosa were mucins, containing large glycoprotein multiple O-glycans. N-acetylgalactosanimyltransferases (GalNAc-Ts) catalyze the transfer of *N*-acetylgalactosamine (GalNAc) to serine/threonine residues during mucin-type O-glycosylation [5].

We also selected the TGF-β signaling pathway as a representative mechanism for CRS development according to nasal polyps. We predicted that six miRNA (miR-1290, miR-548q, miR-548l, miR-1254, miR-890, and miR-548aa/548t) that target genes, such as TGF-β, SMAD, and MAPK, are associated with CRS development. Previous evidence has shown that activated TGF-βR induces phosphorylation of the pathway-restricted SMAD that binds to SMAD. TGF-β and SMAD are significantly increased in the airway mucosa of asthmatic lungs and are associated with thickening of the basement membrane, increased number of fibroblasts, and severity of the disease [12]. These SMAD pathways are not the only means by which TGF-βs regulate cellular functions. SMAD-independent pathways, including the MAPK, nuclear factor κ-light chain-enhancer of activated B cells (*NF*-κ*B*), and PI3 kinase/AKT pathways, also participate in TGF-β signaling. These pathways can either be induced by TGF-β or modulate the outcome of TGF-β-induced SMAD signaling. *MAPK* is a pathway acting downstream of the TGF-β signaling pathway, wherein TGF-β1 activates the MAPK signaling pathway and is associated with myofibroblast differentiation and extracellular matrix production. Furthermore, MAPK induces protein expression associated with tissue remodeling (alpha-smooth muscle actin (a-SMA), fibronectin, and collagen type I) in nasal polyposis [47].

On the basis of these results, we believe that miRNAs that regulate mucin-type O-glycan biosynthesis and the TGF-β signaling pathway may have important functions during CRS progression and aggravation to CRSwNP.

Compared to mRNA and protein, miRNA is relatively stable, making it a potential candidate as a noninvasive biomarker. Some preclinical studies on cancer have already offered interesting possibilities for miRNAs to serve as biomarkers for disease diagnosis, stratification, and monitoring [48,49].

Currently, in vivo studies on the therapeutic effects of miRNAs in inflammatory upper respiratory diseases are lacking. We will study in-depth the mechanism of miRNAs that are regulated in EVs during CRS development in the future. Undoubtedly, translational studies of the clinical use of miRNAs in inflammatory upper respiratory diseases will finally change our approach to the diagnosis, prognosis, and treatment of CRS.

## 5. Conclusions

This study is the first to report on the feasibility of research aimed at determining the mechanisms of disease development involved in the presence of miRNA-carrying EVs in NLF. The ability of miRNAs to transfer to the disease site and their chronicity after reaching the intended location are all aspects of intense development of CRS diagnostics and treatment. There may be significant dysregulated EV miRNAs in CRS patients, and understanding them will be beneficial for future clinical interventions.

## Figures and Tables

**Figure 1 biomedicines-09-00471-f001:**
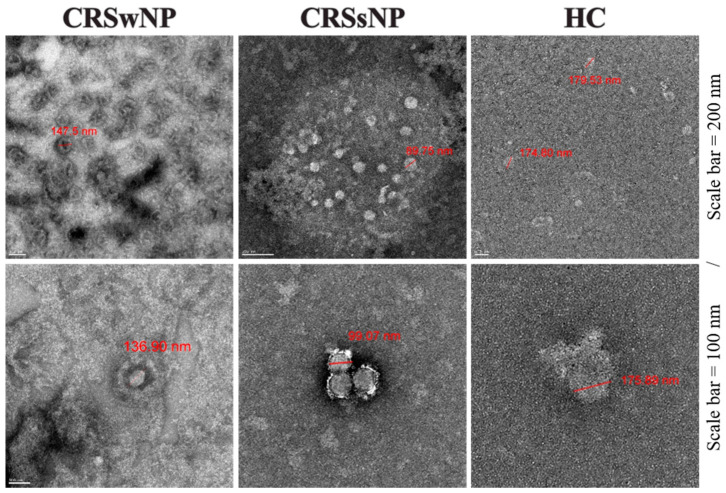
Phenotypic characterization of NLF-EVs. EV shape and size were analyzed by TEM. NLF, nasal lavage fluid; EV, extracellular vesicle; TEM, transmission electron microscopy.

**Figure 2 biomedicines-09-00471-f002:**
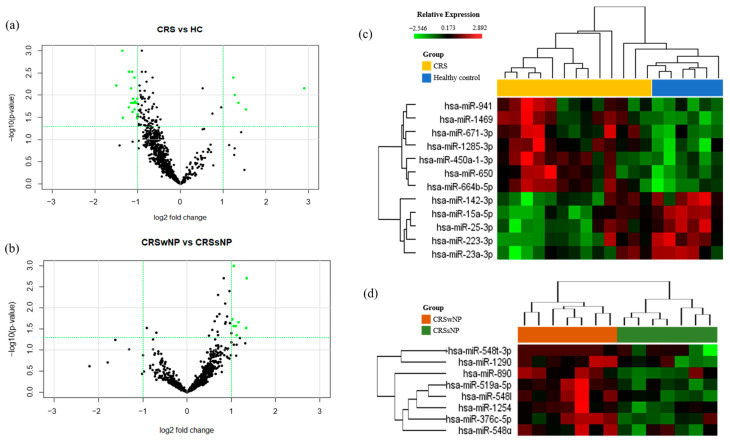
Microarray and analysis of the results in CRS and healthy NLF-EVs. (**A**) Volcano plot showing the relationship between FC and statistical significance. The point of the plot indicates a differentially expressed miRNA with statistical significance. (**B**) Heat map diagram showing cluster analysis of differently expressed miRNAs. Heat maps and hierarchical clusters of selected differentially expressed miRNAs (*p* < 0.05). The sample clustering tree appears at the top, and the miRNA clustering tree is on the left. The color scale reflects the log2 signal strength and goes from green (low intensity) to black (medium intensity) to red (high intensity). The dendrogram at the top and left reflects the hierarchical similarity of the sample and miRNA individually. CRS, chronic rhinosinusitis; NLF, nasal lavage fluid; EV, extracellular vesicle; FC, fold change; miRNA, microRNA.

**Figure 3 biomedicines-09-00471-f003:**
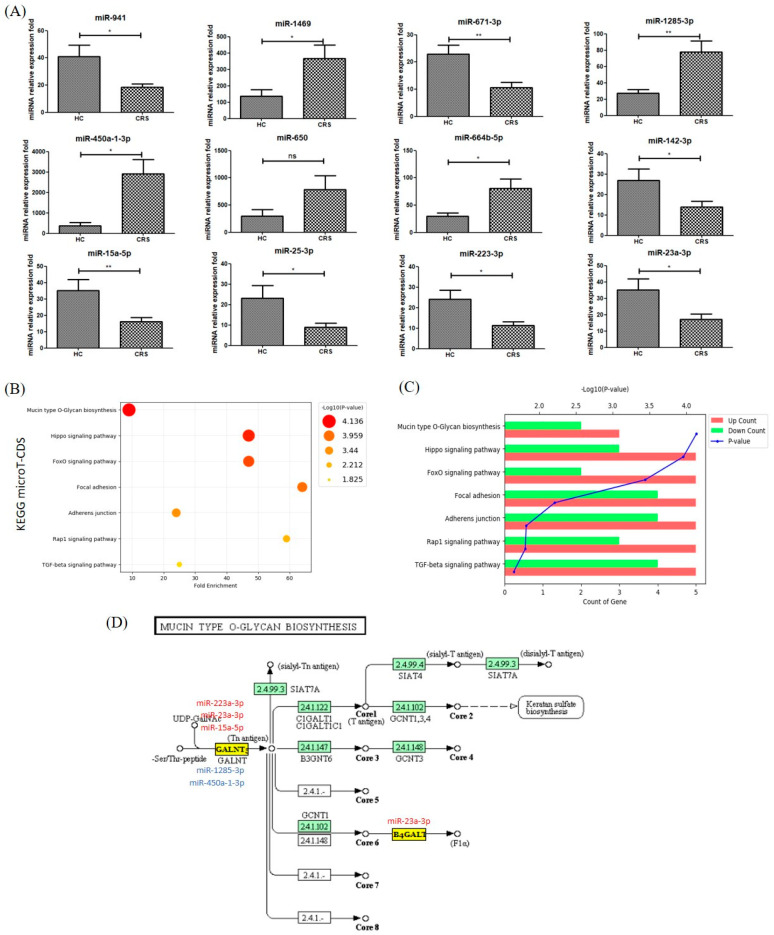
Top 7 KEGG biological pathways revealed by analyzing target miRNAs between NLF-EVs of CRS patients and HCs using DIANA tools. (**A**) Overexpressed miRNAs in NLF-EVs of CRS patients compared to HCs after adjustment of the outlier values according to the Chauvenet criterion (Student’s *t*-test; * *p* < 0.05; ** *p* < 0.01, ns > 0.05). (**B**) KEGG enrichment analysis of all DEGs with |FC| > 2 (cutoff). (**C**) Pathway enriched by up- and downregulated genes and sorted by the value of −log10(*p*-value). (**D**) Diagram of the mucin-type O-glycan biosynthesis. Five dysregulated miRNAs (miR-223-3p, miR-23a-3p, miR-15a-5p, miR-1285-3p, and miR-450a-1-3p) were associated with the mucin-type O-glycan biosynthesis by possibly regulating their potential gene targets. KEGG, Kyoto Encyclopedia Gene and Genome; miRNA, microRNA; NLF, nasal lavage fluid; EV, extracellular vesicle; CRS, chronic rhinosinusitis; HC, healthy control; DEG, differentially expressed gene; FC, fold change.

**Figure 4 biomedicines-09-00471-f004:**
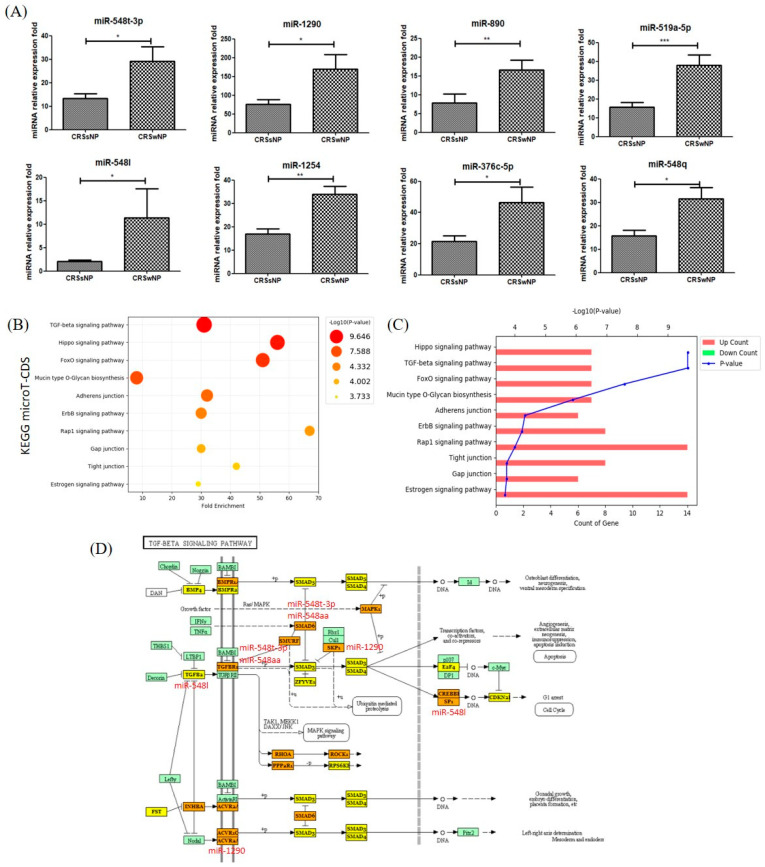
Top 10 KEGG biological pathways revealed by analyzing the target miRNAs between CRSsNP and CRSwNP NLF-EVs using DIANA tools. (**A**) Overexpressed miRNAs in NLF-EVs of CRS patients compared to HCs after adjustment of outlier values according to the Chauvenet criterion (Student’s t-test; * *p* < 0.05; ** *p* < 0.01; *** *p* < 0.001). (**B**) KEGG enrichment analysis of all DEGs with |FC| > 2 (cutoff). (**C**) Pathway enriched by up- and downregulated genes and sorted by the value of −log10(*p*-value). (**D**) Diagram of the TGF-β signaling pathway. Six dysregulated miRNAs (miR-1290, miR-548q, miR-548l, miR-1254, miR-890, and miR-548aa/548t) were associated with the TGF-β signaling pathway by possibly regulating their potential gene targets. KEGG, Kyoto Encyclopedia Gene and Genome; miRNA, microRNA; CRSsNP, chronic rhinosinusitis without nasal polyps; CRSwNP, chronic rhinosinusitis with nasal polyps; NLF, nasal lavage fluid; EV, extracellular vesicle; CRS, chronic rhinosinusitis; HC, healthy control; DEG, differentially expressed gene; FC, fold change; TGF-β, transforming growth factor-beta.

**Table 1 biomedicines-09-00471-t001:** Demographics of CRS patients and HCs for microarray.

Variables	CRSwNP ^1^ (*n* = 7)	CRSsNP ^2^ (*n* = 8)	HC ^3^ (*n* = 7)
Sex, male (%)	3 (42.9%)	4 (50.0%)	6 (85.7%)
Age, years	47.4 ± 23.2	42.0 ± 20.6	40.3 ± 15.2
JESREC score ^4^	9.67 ± 3.27	7.29 ± 5.25	NA
Allergy			
(+)	4 (57.1%)	3 (37.5%)	2 (28.6%)
(−)	3 (42.9%)	5 (62.5%)	5 (71.4%)
Asthma			
(+)	7 (100%)	7 (100%)	7 (100%)
(−)	0 (0%)	0 (0%)	0 (0%)

^1^ CRSwNP, Chronic rhinosinusitis with nasal polyps patients; ^2^ CRSsNP, Chronic rhinosinusitis without nasal polyps patients; ^3^ CRSwNP, Chronic rhinosinusitis with nasal polyps patients; ^4^ JESREC score, the scoring system for ECRS patient classification that it is suitable for the degree of peripheral blood eosinophilia in East Asian patients [6].

**Table 2 biomedicines-09-00471-t002:** EV size were analyzed by transmission electron microscopy (TEM).

Subject	Median Size (IQR) (nm)	Mean Count (Cell Count/10 mm^2^ ± SD)
CRSwNP ^1^	131.56 (107.99–166.29)	75 ± 19
CRSsNP ^2^	82.32 (75.43–102.77)	66 ± 20
HC ^3^	183.93 (157.14–216.07)	3 ± 3
**Total**	**124.57 (93.78–157.14)**	**N/A**

^1^ CRSwNP, Chronic rhinosinusitis with nasal polyps patients; ^2^ CRSsNP, Chronic rhinosinusitis without nasal polyps patients; ^3^ CRSwNP, Chronic rhinosinusitis with nasal polyps patients.

**Table 3 biomedicines-09-00471-t003:** Functional validation of representative miRNA analysis results.

KEGG Pathway	MiRNA	CRS vs. HC	CRSwNP vs. CRSsNP	Target MRNA Name	Putative Disease Associations (Human Studies)
Mucin type O-Glycan biosynthesis	miR-23a-3p	Up	*p* > 0.05	MET; CTNND1; LMO7; IQGAP1; EGFR; TJP1; NLK; CDH1; SMAD4; CTNNB1; CTNNA1; WASF2; CSNK2A1; FARP2; PTPRJ; ACTN4; MAPK1; TGFBR2	Fibrosis [12], Disease Progression [13], Folic Acid Deficiency [14], Genetic Predisposition to Diseases [15]
	miR-15a-5p	Up	*p* > 0.05	ACTB; SMAD2; PVRL2; ACTG1; LMO7; IQGAP1; IGF1R; PTPRF; ACP1; CDH1; CTNNB1; CSNK2A1; RAC1; CDC42; EP300; YES1; FGFR1; MAP3K7; CREBBP; PVRL1	Disease Progression [16]
	miR-223-3p	Up	*p* > 0.05	PARP1; SCARB1; RASGRP1; DDIT4; GFPT1; RGS1; MECP2; MT1E; TTBK2; WDR7; FADS1; STMN1; GDI1; CHUK; SOX5; SERINC1; TMEM69; CPEB4; RNF213; ARTN; COX7A2L	Inflammation [17,18] *, Disease Progression [19,20] *, Genetic Predisposition to Disease [21], primary myelofibrosis [12,22]
	miR-1285-3p	Down	*p* > 0.05	WBSCR17	Inflammation [23]
TGF-beta signaling pathway	miR-548q	*p* > 0.05	Up	IGF1R	Disease Progression [24,25] *, Genetic Predisposition to Disease [26,27], Inflammation [28] *
	miR-1290	*p* > 0.05	Up	CDKN2B; SKP1; ACVR2A; SP1; MAPK1; BMPR2; RPS6KB1	Inflammation [29]
	miR-548l	*p* > 0.05	Up	FST; SMAD2; INHBB; SKP1; ACVR2B; SMAD4; E2F5; SMAD5; ACVR2A; SP1; ACVR1C; TGFB2; BMPR1A; E2F4; PPP2R1B	Acute diseases, Acute Lung Injury [30], Inflammation [31]

## Data Availability

The authors confirm that the data supporting the findings of this study are available within the article and Appendix A.

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
