# Peer review of "MicroRNA Expression in Extracellular Vesicles from Nasal Lavage Fluid in Chronic Rhinosinusitis"

_biomedicines, 2021, doi:10.3390/biomedicines9050471_

Round 1
Reviewer 1 Report
This study is about an interesting field of research, the biological pathways involved in CRS development and in the presence or absence of polyps in CRS. The authors investigated the microRNA expression in extracellular vesicles from nasal lavage fluid and the miRNAs role in disease progression, inflammation, and acute disease. They compare their findings with the results of other published studies and comment on the discrepancies of the published relevant literature to date. They focus on miRNAs that regulate mucin-type O-glycan biosynthesis and the TGF-β signaling pathway that may have important functions during CRS progression and aggravation to CRSwNP. This line of research is contributing toward understanding of the various inflammatory profiles of CRS patients and may ultimately enable individualized treatment protocols. Translational studies of the of miRNAs role in inflammatory upper respiratory diseases may inform our approach to the diagnosis, prognosis estimation, and treatment of CRS.
Specific remarks
Unfortunately lines are not numbered in the manuscript- a highlighted version is attached to help identify the commented text lines
Page 1
Abstract
All abbrevations need to appear first in complete words in the abstract as well as in the text
2nd line of abstract: microRNA (miRNA)
3rd line of abstract: investigated instead of determined
5th line of abstract: Transmission electron microscopy (TEM)
5th line of abstract: KEGG in full ()
9th line of abstract: HCs in full ()
12th line of abstract: TGF-β in full ()
Page 2
Line 8: relative to…. compared to
Line 14: “This study determined the presence of EVs in the NLF of CRS patients compared to HCs” better… “This study investigated the presence of EVs in the NLF of CRS patients and compared it to HCs”.
Page 4
Results_3.2
guidelines of the International Society of Extracellular Vesicles….reference to published guidelines
Page 5
“3.3. MicroRNAs were differentially expressed in NLF-EVs of CRS patients.” To be deleted from Figure 2 description
Page 7
Figure 4. ….. biological pathways
3.4 first paragraph
“miR-1285-3p (FCmiR-1285-3p = 2.85; P = 0.01) significantly increased, while miR-671-3p and miR-15a-5p (FCmiR671-3p = −2.19; P = 0.01 and FCmiR-15a-5p = −2.17; P = 0.001) significantly decreased in individual NLF-EVs of CRS patients compared to HCs” to change to
“In individual NLF-EVs of CRS patients, miR-1285-3p (FCmiR-1285-3p = 2.85; P = 0.01) significantly increased, while miR-671-3p and miR-15a-5p (FCmiR671-3p = −2.19; P = 0.01 and FCmiR-15a-5p = −2.17; P = 0.001) significantly decreased compared to HCs”
Discussion - Page 10 - last paragraph
“However, in recent reports, as “One Airway, One Disease,” only the sites of occurrence are different and Huse et al. are regarded as the same disease, such that nasal diseases and respiratory diseases are considered the same disease [40].”
Rephrase and present with clarity the associations of the concept of “One Airway, One Disease” to Huse et al., reference 40 and the findings of the present study.
Page 11 - second paragraph
“We performed a pilot study of…”
“…using tissue samples from 8 patients who obtained NLF EV samples…” can be rephrased to
“…using tissue samples from 8 patients, whom NLF EV samples were obtained from…” can be rephrased to
“In the same group comparison as in the Xuan’s and Xia’s studies” or “In the same comparison of groups as in the Xuan’s and Xia’s studies”
Either SMAD or Smad consistently

Author Response
We thank the reviewers for their constructive criticism, which helped us to improve the manuscript. We have attempted to carefully and thoroughly address all the reviewers’ concerns. The response was marked in red. And the changes in the revised manuscript are indicated in red and bold font for easy identification. Also, all of the changes are made equally using the “Track Changes” function in the manuscript file (Microsoft Word) submitted together.
Please see the attachment.

Reviewer 2 Report
I appreciate the opportunity to review the manuscript for publication in MDPI Biomedicines.
The authors examined the presence of EVs in the nasal lavage in CRS patients, and clearly demonstrated different miRNA profiles related certain biological pathways stratified by CRS phenotypes.
I feel that the topics is interesting and well organized.
I have a few minor comments.
Table 1: I have never heard “j-score”. The authors should cite the Ref. 7 in a correct manner.
Supplementary Table S1: The definition of ECRS is inaccurate. ECRS should be included in the CRSwNP category. Any patients w/o NP cannot be included.
Supplementary Fig. S1C: Luminosity of each fluorescence is less qualified for presentation.
Author Response

(The authors gave the same response as above.)
